# Primary Sequence-Intrinsic Immune Evasion by Viral Proteins Guides CTL-Based Vaccine Strategies

**DOI:** 10.3390/v17081035

**Published:** 2025-07-24

**Authors:** Li Wan, Masahiro Shuda, Yuan Chang, Patrick S. Moore

**Affiliations:** 1Cancer Virology Program, Hillman Cancer Center, University of Pittsburgh, Pittsburgh, PA 15213, USA; liw113@pitt.edu (L.W.); mas253@pitt.edu (M.S.); 2Department of Pathology, University of Pittsburgh, Pittsburgh, PA 15213, USA

**Keywords:** primary sequence-intrinsic immune evasion, MHC I, CTL-based vaccine

## Abstract

Viruses use a range of sophisticated strategies to evade detection by cytotoxic T-lymphocytes (CTLs) within host cells. Beyond elaborating dedicated viral proteins that disrupt the MHC class I antigen-presentation machinery, some viruses possess intrinsic, cis-acting genome-encoded elements that interfere with antigen processing and display. These protein features, including G-quadruplex motifs, repetitive peptide sequences, and rare-codon usage, counterintuitively limit production of proteins critical to virus survival, particularly during latency. By slowing viral protein synthesis, these features reduce antigen production and proteosomal degradation, ultimately limiting the generation of peptides for MHC I presentation. These built-in evasion tactics enable viruses to remain “invisible” to CTLs during latency. While these primary sequence intrinsic immune evasion (PSI) mechanisms are well-described in select herpesviruses, emerging evidence suggests that they may also play a critical role in RNA viruses. How these proteins are made, rather than what they functionally target, determines their immune evasion properties. Understanding PSI mechanisms could rationally inform the design of engineered viral antigens with altered or removed evasion elements to restore antigen CTL priming and activation. Such vaccine strategies have the potential to enhance immune recognition, improve clearance of chronically infected cells, and contribute to the treatment of persistent viral infections and virus-associated cancers.

## 1. Introduction

Viral latency is a phase in which certain viruses persist within host cells in a transcriptionally restricted or dormant state, avoiding active replication and, consequently, detection by the host immune system [1]. While well-characterized in herpesviruses and retroviruses, latency likely occurs in other viral classes capable of establishing prolonged or life-long chronic infections. More precisely, latency is defined as a state in which viral genome replication is strictly coupled to the host cell, rather than proceeding autonomously [2]. The mechanisms of latency vary by virus and host cell type. For example, in herpes simplex virus (HSV) latency, the viral genome persists in post-mitotic G_0_-phase neurons, where neither the virus nor the host cell replicates. Here, the expression of a noncoding latency-associated transcript (LAT) is sufficient to maintain viral genome persistence. In contrast, Epstein–Barr virus (EBV) establishes latency in B cells, wherein each viral genome replicates once and only once with each cell cycle—and then partitions between daughter cells as the host cell divides. On transition to the lytic replication phase, the virus genome template replicates multiple times within a single host cell cycle, leading to exponential virus amplification and generally causing death of the host cell.

During latency, viral antigen expression is minimized, allowing the virus to evade detection and immune clearance [3]. Nonetheless, certain latent viruses, including EBV and Kaposi sarcoma herpesvirus (KSHV), must still express a limited set of viral products, including proteins, noncoding RNAs (e.g., miRNAs [4], circRNAs [5]), and regulatory mRNAs to ensure the maintenance and faithful partitioning of latent genomes during cell division. These latent proteins, particularly EBV’s EBNA1 and KSHV’s LANA, have evolved specialized mechanisms to evade immune detection. These mechanisms are encoded directly within the primary sequence of the protein-coding regions and represent a distinct class of immune evasion we call primary sequence-intrinsic immune evasion (PSI). This review focuses on PSI as a viral strategy for immune evasion, highlighting its functional importance during latency. Emerging evidence suggests that PSI is not limited to a few herpesviruses but may be a broader strategy employed by a range of viral pathogens, including RNA viruses [6].

Central to host defense against intracellular pathogens, particularly viruses, is the MHC class I antigen presentation pathway, which enables cytotoxic T lymphocytes (CTLs) to detect and eliminate infected cells [7]. Proteins synthesized in cells, either through the standard protein synthesis pathway or as “defective ribosomal products” (DRiPs) [8,9], are processed by the proteasome into short peptides. These peptides are transported into the endoplasmic reticulum (ER) by transporter-associated with antigen processing (TAP) proteins and loaded onto MHC class I molecules. Chaperones assist in stabilizing the peptide-MHC I-β_2_ microglobulin complex [10] which is subsequently transported to the cell surface for presentation to CD8+ cytotoxic T lymphocytes (CTLs). Antigen-presenting cells (APCs) initiate the initial priming of a CTL immune response to a peptide antigen. Recognition of these complexes by the T cell receptor (TCR) on CD8+ CTLs triggers effector functions, including the release of cytokines and cytotoxins, such as interferon-gamma (IFN-γ), and cytotoxic molecules such as perforin and granzyme, which induce apoptosis of the infected cell [11]. Because nearly all nucleated cells are capable of presenting antigens via MHC class I [12], immune evasion at this checkpoint is crucial for viruses to establish and maintain persistent infections, particularly during latency, when viral visibility must remain minimal [13].

## 2. Trans and Cis-Acting MHC Class I-Mediated CTL Immune Evasion

Viruses across all genome types have evolved proteins that disrupt host antigen presentation via MHC class I [14,15,16]. These viral proteins target nearly every step of the MHC I pathway: blocking peptide transport from the cytosol into the endoplasmic reticulum (ER) by interfering with TAP proteins, impairing peptide loading by destabilizing the peptide loading complex (PLC) or preventing MHC I trafficking to the cell surface [14]. Mechanistically, these proteins may bind directly to host factors to inhibit their function, trigger their ubiquitination and proteasomal degradation, or suppress transcription of MHC I regulatory genes.

In addition to these trans-acting strategies, some viruses have evolved cis-acting genetic elements, such as sequence motifs or structural RNA features, that intrinsically impair cis antigen presentation from specific viral proteins. Cis-acting evasion tactics are particularly common in viruses that establish persistent infections. This review focuses on these cis-acting mechanism by which viral proteins subvert MHC class I-mediated CTL responses.

## 3. Viral Sequence-Dependent Intrinsic CTL Immune Evasion

### 3.1. G-Quadruplexes Regulate Viral Protein Expression and Antigen Presentation

G-quadruplexes (G4) are non-canonical nucleic acid secondary structures formed by guanine-rich sequences in DNA or RNA through Hoogsteen hydrogen bonding, rather than standard Watson–Crick double-stranded base-pairing [17]. They form in guanine-rich regions of DNA or RNA when four guanine bases align to create planar G-tetrads, which can stack on one another and are stabilized by π–π interactions in combination with a sodium ion, resulting in a stable single-stranded G4 conformation [18]. In eukaryotic systems, G4 structures are known to play important regulatory roles in both DNA and RNA, where they modulate key processes such as DNA replication, transcriptional initiation and elongation, RNA splicing, translation, and the activity of noncoding RNAs [19].

Notably, G4 structures are also widespread and highly conserved across the genomes of many human viruses [20]. Their presence was first predicted through in silico analysis [21] with subsequent confirmation by in vitro biochemical assays [22]. In promoter regions, viral G4s have been well studied for their role in regulating viral replication and transcription. For example, in herpes simplex virus-1 (HSV-1), G4s located in the promoters of immediate-early genes *ICP0* and *ICP4* interfere with transcription factor binding and RNA polymerase access, thereby repressing gene expression [23]. Emerging evidence highlights a critical role for G4s within viral open reading frames (ORFs), where they contribute to post-transcriptional regulation and PSI. For instance, EBV encodes the EBNA1 protein, whose mRNA contains G4-rich structures within its glycine–alanine (GAr) repeat domain. These G4s limit translation efficiency, reducing protein output and subsequence MHC class I antigen presentation (Figure 1) [24]. Disruption of these G4s with antisense oligonucleotides enhances EBNA1 protein translation and increases antigen presentation on MHC class I [25], while stabilization has the opposite effect, suppressing both translation and antigen visibility [25].

A similar mechanism occurs in KSHV. The mRNA encoding the latency-associated nuclear antigen (LANA) contains G4 in its central repeat domain, particularly within the Central Repeat (CR) 3 region (a.a. 857–917), as demonstrated by circular-dichroism spectroscopy [26]. Pharmacologic stabilization of these G4s using the small molecule TMPyP4 results in reduced LANA translation and decreased antigen presentation [26]. LANA itself can bind its own G4-containing mRNA, suppress its cytoplasmic export and further limit cytoplasmic translation. Intriguingly, although the genes encoding LANA (*ORF73*) and EBNA1 (*BKRF1*) are highly homologous, their protein products differ significantly due to +1 frameshift recoding events that may arise from ribosomal stalling at G4 [27]. This mechanism expands the coding potential of the viral genome and may contribute further to immune evasion.

The prevailing model suggests that RNA G4s form stable secondary structures that physically impede ribosome elongation. This ribosomal stalling reduces overall translation and downstream antigen presentation via MHC class I [28,29]. Additional layers of regulation come from host G4-binding proteins such as nucleolin, which binds to the G4 regions of LANA mRNAs and retains them in the nucleus. Knockdown of nucleolin via shRNA relocalizes the mRNA to the cytoplasm and significantly increases LANA protein levels [30]. Similarly, nucleolin binds EBNA1 GAr RNA at its G4 structure, further reducing its translation and MHC I presentation (Figure 1) [31]. By encoding G4 elements directly in their primary genome sequences, latent viruses like EBV and KSHV suppress the immunogenicity of key latency-associated proteins. This strategy limits antigen availability for CTL recognition, promoting viral persistence and immune evasion [32,33].

### 3.2. Resistance to Proteasomal Degradation

Human herpesvirus KSHV and EBV latent proteins, as well as latent proteins from related animal herpesviruses, possess repetitive peptide domains that impede protein translation to promote correct folding and reduce DRiP presentation [34,35]. EBV’s principal latent antigen EBNA1 is exceptionally stable, limiting its proteasomal processing [36]. Its Gly-Ala repeat (GAr) region tethers EBNA1 to chromatin via AT-hook-like motifs, an interaction that does not block ubiquitylation but impedes the proteasome’s ability to extract the substrate (Figure 1) [37,38,39]. AT-hook of HMGA protein adopts a crescent-like structure that fits into the minor groove of DNA, where arginine side chains interact hydrophobically with adenines to lock the domain in place [40]. Although the EBNA1 GAr lacks the typical proline residue and tripartite organization of the AT-hook, its extended Arg-Gly-Arg sequences likely strengthen DNA binding by engaging multiple AT base pairs simultaneously [37]. EBNA1’s localization in the nucleus also restricts its access to the macroautophagy pathway, thereby reducing both the amount and diversity of EBNA1-derived CD4+ T cell epitopes presented on the surface of infected cells [41,42]. Despite its nuclear localization, EBNA1 is detected by CD4+ T cells in EBV-positive lymphoma cells [43] and is presented on MHC class II molecules via macroautophagy [44]. Therefore, regardless of the MHC molecules exploited for EBNA1 presentation, PSI-mediated reduction in the quantity and diversity of EBNA1-derived TCR epitopes could be crucial for evading immune responses from both CD4+ and CD8+ T cells. EBNA1 resists degradation in cis through GAr-mediated disruption of its engagement with the 19S proteasomal subunit [45,46]. EBNA1-derived DRiPs lacking GAr can still be processed and presented on MHC I [47,48]. Moreover, fusion of EBV’s GAr to p53 protects the chimera from Mdm2- and human papillomavirus (HPV)-E6-induced degradation, underscoring the sequence-encoded stability conferred by GAr [49].

Similarly, the central repeat region of KSHV LANA, rich in glutamine, glutamate, and aspartate residues, has been found to confer proteasomal resistance, resulting in an extremely long LANA protein half-life [50]. When fused to heterologous proteins, this region likewise stabilizes them and reduces their degradation and immunogenicity in cis [50]. Unlike EBNA1, LANA lacks an AT-hook–like motif. However, it interacts with histones through its N-terminal domain [51], which may similarly aid in resisting proteasomal degradation and promoting immune evasion. Although LANA is highly immunogenic in inducing humoral antibody responses [52], CTL responses to LANA peptides are absent from KSHV-positive persons [53] suggesting that these features prevent antigen priming of an anti-LANA CTL response.

The related herpesvirus saimiri (HVS) establishes long-term infection in squirrel monkeys and has an ORF73 gene homologous to the KSHV LANA-encoding gene that generates a protein with a central region of amino acid repeats [54]. The HVS ORF73 repeat region is comprised of repetitive glutamic acid–glycine (EG) segments linked to glutamic acid-alanine (EA) segments. These EG-EA repeats also serve to reduce CD8+ CTL recognition of ORF73 [55]. Unlike EBNA1 or LANA, deletion of this repeat domain does not affect the protein’s stability or translation rate in in vitro translation assays; instead, the EEAEEAEEE motif lowers ORF73 mRNA levels, thereby reducing CTL recognition through an unknown mechanism [55]. Alcelaphine herpesvirus 1 (AlHV-1) infects wildebeests persistently and asymptomatically but causes malignant catarrhal fever (MCF) when transmitted to other ruminants [56]. AlHV-1 ORF73 encodes the latency-associated nuclear antigen (LANA)-homolog protein (aLANA). aLANA contains a glycine/glutamate (GE)-rich repeat that suppresses its own MHC I presentation. This GE repeat does not affect aLANA protein turnover, but slows translation via its mRNA sequence—an effect reversed by codon modification, which significantly enhances MHC I presentation in vitro and CTL response in vivo [56].

### 3.3. Rare Codon Usage

Synonymous codon choices are not equivalent: organisms exhibit codon usage bias, favoring certain codons with more abundant cognate tRNAs [57]. Rare codons can slow translation by causing ribosomal pausing when their tRNAs are scarce [58,59]. This bias arises from a balance of mutational pressures and natural selection for translational efficiency [60], and viral codon usage often reflects, but does not precisely mirror, host preferences [61]. In latent infections, viruses may exploit rare codons to throttle antigen synthesis and avoid immune detection. Human cytomegalovirus (HCMV), for example, frequently uses six rare codons: GCG (Ala), CCG (Pro), CGT (Arg), CGC (Arg), TCG (Ser), and ACG (Thr), which reduce translation rates [62]. Hepatitis A virus (HAV) similarly eschews abundant host codons to limit protein output [63,64]. In EBV, codon usage for lytic genes aligns closely with the host, whereas latent-phase genes diverge markedly, supporting a strategy of low-level expression for persistent infections [65]. By constraining antigen synthesis, rare-codon usage diminishes peptide availability for MHC I loading without compromising essential viral functions (Figure 1).

### 3.4. Intrinsic Immune Evasion in RNA Viruses

Although in cis immune evasion is often associated with latent DNA viruses, RNA viruses also evolve PSI mechanisms to escape CTL surveillance. With smaller genomes and high mutation rates, RNA viruses can rapidly adapt their coding sequences to evade T cell recognition [66].

Influenza A, under T cell pressure, acquires mutations not only within epitopes but also in flanking regions, impairing antigen recognition [67]. As a result, CTLs specific to human influenza viruses primarily target epitopes found in highly conserved internal viral proteins, such as M1, NP, PA, and PB2 [68,69]. However, there is clinical evidence that CTL contributes significantly to the immune surveillance of these viruses in humans [69]. A recent study showed that influenza A generates an alternative reading frame DRiP epitope functionally stimulating CTL response [70], which may suggest that unexpected ribosomal stalling may produce misfolded protein with enhanced proteasomal degradation and MHC I presentation.

SARS-CoV-2 vaccines primarily target the spike protein, which is known for its high mutation rate [71]. The non-structural replication proteins of SARS-CoV-2 are highly conserved and could be ideal for generating broad protection across variants. SARS-CoV-2 RNA-dependent RNA polymerase (RdRp or Nsp12) may offer broader protection, with >99% amino acid sequence identity among all known strains [72,73] and 96% and 71% identity to SARS-CoV and MERS-CoV RdRp proteins, respectively [6]. A recent study from our group found that the SARS-CoV-2 RdRp minimizes its synthesis and MHC I presentation through an unclear PSI mechanism that may rely on protein folding or post-translational modifications (e.g., nucleotidylyzation) to prevent immune recognition [6]. Full-length codon-optimized SARS-CoV-2 RdRp has minimal MHC I presentation compared to EGFP or KSHV LANA [6], making it an impractical vaccine target if produced directly from mRNA based on the wild-type or codon-optimized viral sequence. However, if the RdRp finger domain is deleted from the open reading frame and expressed in trans with the edited RdRp, so that these two fragments represent almost all peptide epitopes for RdRp (so-called RdRp^Frag^), potent CTL priming and recognition are restored when administered to C57Bl/6 mice [6]. This illustrates that simple protein engineering can overcome PSI in viral proteins. It is unknown whether this will generate a clinically relevant pan-strain SARS-CoV-2 vaccine since RdRp is scantily expressed during COVID infection [74] but it does suggest that PSI plays a broader role than previously suspected for non-herpesvirus infections and points towards a strategy for maximizing sub-unit vaccine CTL responses.

## 4. Enhanced CTL Vaccines by Reversing Intrinsic Immune Evasion

CTL-based vaccines, including mRNA vaccines, live-attenuated vaccines, and recombinant vector vaccines, aim to bolster cellular immunity by delivering viral antigens that engage the MHC I pathway, thereby activating and expanding antigen-specific CD8+ [75]. By targeting antigen-presenting cells (APCs), these vaccines ensure efficient peptide loading onto MHC I and subsequent presentation to CTLs [76]. Unlike subunit, toxoid, or inactivated vaccines that typically elicit B cell-mediated antibody responses [77], CTL-based vaccines directly target infected cells and are especially valuable against pathogens that evade neutralizing antibodies. For example, HIV vaccine studies have shown that robust CTL responses can reduce viral loads and slow disease progression by recognizing and destroying infected cells, even when the virus mutates to escape antibody recognition [78,79].

A key challenge in CTL vaccine design is the use of viral subunit antigens that, by virtue of PSI mechanisms, are poorly presented on MHC I (i.e., “cold proteins”) and thus fail to elicit strong CTL responses. By understanding and reversing these evasion strategies, however, it is possible to transform cold proteins into hot antigens while preserving essential T-cell epitopes. For example, deletion of the CR1 domain of KSHV LANA restores its MHC I presentation in vitro [80]. A CTL-based vaccine using CR1 deleted LANA mRNA or peptide could therefore enhance presentation of LANA-derived peptides and stimulate LANA-specific CTLs—responses that natural infection typically fails to generate [6,53]. Likewise, removal of EBNA1’s GAr region [45,46] or mutation of the EEAEEAEEE motif in HVS ORF73 [55] can similarly boost antigen presentation and CTL activation.

A potential stumbling block to this approach is that even if deleting PSI motifs markedly augments priming and boosting of CTL responses to an antigen, would this be sufficient to achieve protective surveillance during subsequent viral infections in which the protein is still processed through PSI mechanisms?

The principal method to screen for viral protein PSI utilizes a simple but robust bioassay developed by Yewdell and colleagues [81,82]. In this assay, human 293 cells stably expressing murine H2K^b^ MHC I can be transfected with a DNA encoding the viral protein of interest fused to an eGFP tag (to measure protein expression) and a SIINFEKL epitope tag (derived from chicken ovalbumin). Flow cytometry allows simultaneous quantitation of both eGFP expression and cell surface H2K^b^ MHC I cell surface presentation to determine the presentation ratio for the viral protein (Figure 2). This assay provides a method for screening of engineered antigens to determine presentation efficiency and potential PSI activity. It also allows antigen manipulation by simple genetic engineering to establish the best means for overcoming PSI (e.g., deletion of a PSI domain, generation of a destabilized antigen, etc.). This surprisingly robust and simple bioassay is reproducible across a variety of different transient transfection conditions [6]. By surveying deletions or mutations in a protein of interest, it is possible to map protein regions that suppress MHC I peptide presentation.

Positive hits can then be evaluated for functional CTL by co-culturing SIINFEKL-specific CD8+ T cells (OT-1) with antigen-expressing APCs and measuring cytokine production (e.g., IFNγ, TNFα, etc.) or CTL-mediated killing (Figure 2) [6,83,84]. This workflow enables rapid identification and optimization of vaccine candidates that can effectively reverse primary sequence intrinsic immune evasion to elicit potent CTL responses.

While generating high frequencies of CD8+ T cell responses is desirable, it is equally important to consider the functional quality of the elicited T cells. Effective immune protection depends not only on the quantity of T cells but also on their avidity—a factor determined by both the affinity of the T cell receptor (TCR) and the expression of co-stimulatory receptors such as CD28, 4-1BB, and OX40 [85]. Therefore, vaccine strategies should also aim to enhance T cell avidity, potentially through the use of booster doses with an optimized vaccination schedule [86], as well as targeting co-stimulatory molecules directly [87,88].

## 5. Conclusions

Viral latency and immune evasion represent formidable barriers to effective immune surveillance and intervention. By co-opting multiple strategies ranging from blockade of MHC class I antigen presentation and resistance to proteasomal degradation, to the formation of G-quadruplex-mediated translational roadblocks, and deployment of rare codons, viruses sharply curtail CTL recognition and killing of infected cells. These layered defenses underscore the dynamic co-evolution between host immune systems and viral pathogens. While these mechanisms were first identified in the herpesvirus EBV EBNA1 protein [45] and extended to related herpesvirus proteins, such as KSHV LANA, it is likely that PSI plays a broader role than previously recognized in viral immune evasion.

Deciphering the molecular basis of these evasion mechanisms provides a foundation for innovative therapeutic approaches. Disrupting G-quadruplexes, removal of cis-acting stability motifs, or codon optimization strategies can convert cold viral antigens into highly immunogenic targets that robustly engage MHC I and activate CTLs. Leveraging these insights into next-generation CTL-based vaccines and immunotherapeutics—potentially effective against herpesviruses, persistent RNA viruses, and emerging pathogens such as SARS-CoV-2—promise to elicit durable cellular immunity and, ultimately, to surmount the barriers posed by viral latency.

## Figures and Tables

**Figure 1 viruses-17-01035-f001:**
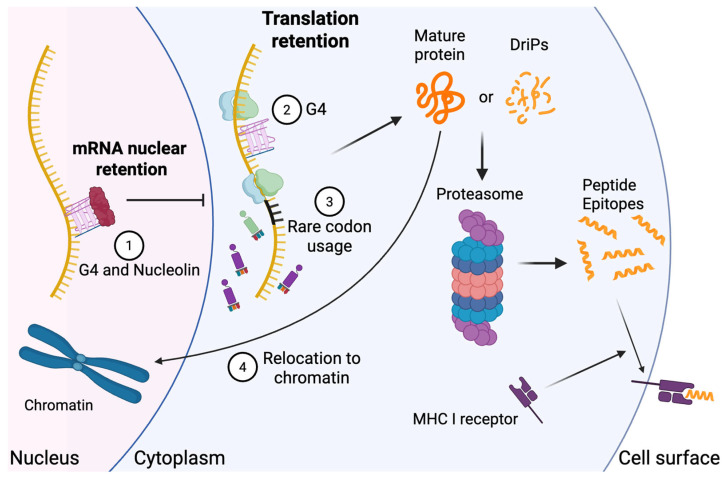
**Primary sequence-intrinsic viral immune evasion (PSI) mechanisms targeting the MHC I presentation pathway**. (**1**) The central repeat (CR) region of KSHV LANA forms G-quadruplex (G4) RNA structures that recruit nucleolin, leading to nuclear of the transcript and reduced cytoplasmic translation, (**2**) G4 structures in the mRNAs of KSHV LANA and EBV EBNA1 impede translation elongation by inducing ribosomal stalling, thereby limiting protein synthesis and promoting proper protein folding, (**3**) Human cytomegalovirus (HCMV), hepatitis A virus (HAV), and EBV exploit rare usage to slow translation by depleting locally available cognate tRNAs, reducing antigen load. (**4**) EBV EBNA1 GAr tethers to chromatin to resist proteasomal degradation.

**Figure 2 viruses-17-01035-f002:**
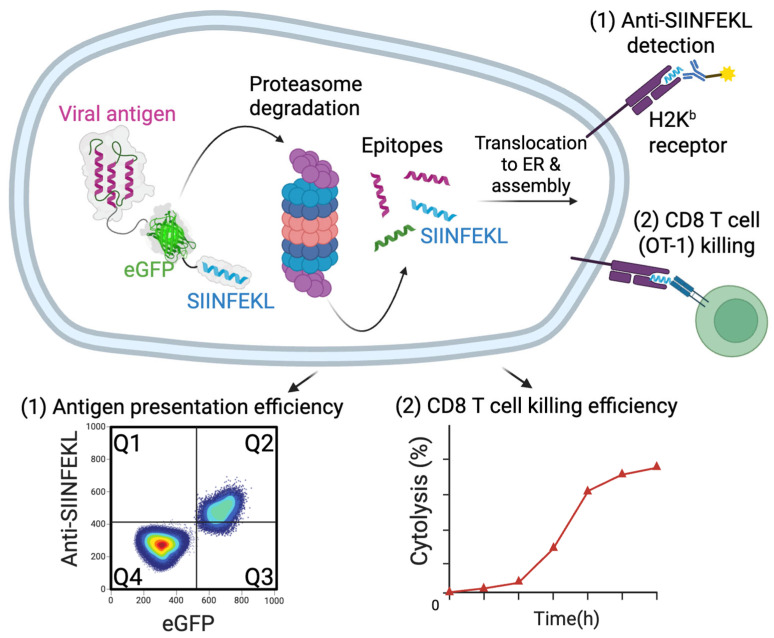
**SIINFEKL-based MHC I antigen presentation assay for evaluating PSI activity.** Viral antigens are fused in-frame to eGFP and the SIINFEKL peptide (derived from chicken ovalbumin) and transfected into H2K^b^-expressing 293 cells. eGFP fluorescence serves as a readout of antigen expression (Q2 + Q3), while SIINFEKL is processed and presented via the endogenous MHC I pathway. Surface presentation of SIINFEKL-H2K^b^ complexes is detected using a conformation-specific anti-SIINFEKL- H2K^b^ monoclonal antibody, enabling quantification of MHC I presentation (Q2). The antigen presentation efficiency is calculated as the ratio, Q2/(Q2 + Q3). This assay can be followed by functional validation using OT-1 CD8+ T cells, which recognize the SIINFEKL-H2K^b^ complex, to assess cytokine production or target cell lysis. Secondary functional screening can be performed with an SIINFEKL-specific tetramer.

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
