# Peer review of "Primary Sequence-Intrinsic Immune Evasion by Viral Proteins Guides CTL-Based Vaccine Strategies"

_viruses, 2025, doi:10.3390/v17081035_

Round 1

Reviewer 1 Report

Comments and Suggestions for Authors

Manuscript #viruses-3729096

Wan et al., “Viral Protein Primary Sequence Immune Evasion Guides CTL-Based Vaccine Strategies

The authors discuss primary sequence immune evasion (PSI) strategies of viruses, including ribosomal inhibition by repeat domains, rare codon usage, secondary RNA structure formation, subcellular compartmentalization and proteasomal inhibition. They propose vaccine design to remove PSIs from viral antigens for vaccination to improve immune control of the respective mostly persisting viral infections.

This is an interesting review on an important topic to improve vaccination for viral pathogens that can be readily addressed with vaccine platforms that have been expanded during the SARS-CoV-2 pandemic. Therefore, the review is timely but should maybe include a discussion on T cell avidity that is required to target infected cells in which PSIs are unchanged.

Major comments:

  1. The PSI of viral protein retention at chromatin shown in figure 1 is only discussed with one sentence in the text. Some sentences for this PSI could be added for the chromatin binding function of EBNA1 and LANA that might serve immune escape similar to GA hooks.
  2. Although high frequencies of CD8+ T cell responses are desirable after vaccination and therefore antigens from which PSIs have been removed should be explored. However, it should be considered that the elicited T cell responses need to have sufficient avidity (mediated both through the T cell receptor affinity as well as co-stimulatory receptor expression due to the optimal T cell differentiation stage) in order to recognize infected targets. Maybe some discussion to this effect could be added.
  3. The authors do not discuss PSIs that cause evasion from CD4+ T cell responses. Maybe some discussion along these lines could be added.

Minor comments:

  1. Line 66: insert “of a” between priming and CTL immune response.

Author Response

Comment 1: The PSI of viral protein retention at chromatin shown in figure 1 is only discussed with one sentence in the text. Some sentences for this PSI could be added for the chromatin binding function of EBNA1 and LANA that might serve immune escape similar to GA hooks.

Reply: We thank the reviewer for comments that help to improve this review.  We include the following text to introduce the AT-hook and EBNA1-DNA binding mechanism. LANA does not have AT-hook motifs but similarly tethers chromatin:

EBV’s principal latent antigen EBNA1 is exceptionally stable which limits its proteasomal processing [36]. The EBNA1 Gly-Ala repeat (GAr) region tethers EBNA1 to chromatin via AT-hook-like motifs, an interaction that does not block ubiquitylation but impedes the proteasome’s ability to extract the substrate (Fig 1) [37-39]. The AT-hook motif of HMGA protein adopts a crescent-like structure that fits into the minor groove of DNA, where arginine side chains interact hydrophobically with adenines to lock the domain into place.[40] Although the EBNA1 GAr lacks a typical Proline residue and tripartite organization of the AT-hook, its extended Arg-Gly-Arg sequences likely strengthen DNA binding by engaging multiple AT base pairs simultaneously [37]. …..

Unlike EBNA1, LANA lacks an AT-hook–like motif. However, it interacts with histones through its N-terminal domain [47], which may similarly aid in resisting proteasomal degradation and promoting immune evasion.

Comment 2: Although high frequencies of CD8+ T cell responses are desirable after vaccination and therefore antigens from which PSIs have been removed should be explored. However, it should be considered that the elicited T cell responses need to have sufficient avidity (mediated both through the T cell receptor affinity as well as co-stimulatory receptor expression due to the optimal T cell differentiation stage) in order to recognize infected targets. Maybe some discussion to this effect could be added.

Reply: The following paragraph is now included in section 4 (Enhanced CTL Vaccines by Reversing Intrinsic Immune Evasion):

While generating CD8⁺ T cell responses are desirable, it is equally important to consider the functional quality of the elicited T cells. Effective immune protection depends not only on the quantity of reactive T cells but also on their avidity—a factor determined by both the affinity of the T cell receptor (TCR) and the expression of co-stimulatory receptors such as CD28, 4-1BB, and OX40 [81]. Therefore, vaccine strategies should aim to enhance T cell avidity, potentially through the use of booster doses with an optimized vaccination schedule [82], as well as by targeting co-stimulatory molecules directly [83, 84]

Comment 3: The authors do not discuss PSIs that cause evasion from CD4+ T cell responses. Maybe some discussion along these lines could be added.

Reply: Some viral antigens were reported to influence MHC II pathway by targeting CIITA (LANA) or interfering other MHC II related proteins such as DR & CD74 (EBNA1, BZLF1, etc.), these mechanisms broadly disrupt MHC II antigen presentation and are not considered PSI. However, EBNA1 was reported to evade MHC II presentation through an intrinsic, antigen-specific mechanism, which we now highlight and expand upon in the revised manuscript:

EBNA1’s localization in the nucleus also restricts its access to the macroautophagy pathway, thereby reducing both the amount and diversity of EBNA1-derived CD4⁺ T cell epitopes presented on the surface of infected cells [41, 42]. Despite its nuclear localization, EBNA1 is detected by CD4+ T cells in EBV-positive lymphoma cells [43] and is presented on MHC class II molecules via macroautophagy [44]. Therefore, regardless of the MHC molecules exploited for EBNA1 presentation, PSI-mediated reduction in the quantity and diversity of EBNA1-derived TCR epitopes could be crucial for evading immune responses from both CD4+ and CD8+ T cells.

Comment 4: Line 66: insert “of a” between priming and CTL immune response.

Reply: Thank you for noting this. We have corrected the sentence as suggested.

Reviewer 2 Report

Comments and Suggestions for Authors

A very precise summary of viral-derived primary sequence-mediated immune evasion, based on the research data from both RNA and DNA viruses. The main evasion discussed in the review paper was CTL-mediated antigen recognition. The key mechanisms, including G4-regulated antigen presentation, the seq repeats-mediated resistance to proteasomal degradation, and rare codon usage, are all well described and discussed. The implication of those mechanisms in vaccine development is well discussed.  

Author Response

Comment 1: A very precise summary of viral-derived primary sequence-mediated immune evasion, based on the research data from both RNA and DNA viruses. The main evasion discussed in the review paper was CTL-mediated antigen recognition. The key mechanisms, including G4-regulated antigen presentation, the seq repeats-mediated resistance to proteasomal degradation, and rare codon usage, are all well described and discussed. The implication of those mechanisms in vaccine development is well discussed.  

Reply: We thank the reviewer for this positive feedback. We are pleased that the manuscript was well received and appreciate your supportive comments.